# Binocular-Guided 3D Gaussian Splatting with View Consistency for Sparse View Synthesis

**Liang Han[1], Junsheng Zhou[1], Yu-Shen Liu[1]***, **Zhizhong Han[2]**

School of Software, Tsinghua University, Beijing, China[1]
Department of Computer Science, Wayne State University, Detroit, USA[2]

`{hanl23,zhou-js24}@mails.tsinghua.edu.cn`
`liuyushen@tsinghua.edu.cn`   `h312h@wayne.edu`

## Abstract

Novel view synthesis from sparse inputs is a vital yet challenging task in 3D computer vision. Previous methods explore 3D Gaussian Splatting with neural priors (e.g. depth priors) as an additional supervision, demonstrating promising quality and efficiency compared to the NeRF based methods. However, the neural priors from 2D pretrained models are often noisy and blurry, which struggle to precisely guide the learning of radiance fields. In this paper, We propose a novel method for synthesizing novel views from sparse views with Gaussian Splatting that does not require external prior as supervision. Our key idea lies in exploring the self-supervisions inherent in the binocular stereo consistency between each pair of binocular images constructed with disparity-guided image warping. To this end, we additionally introduce a Gaussian opacity constraint which regularizes the Gaussian locations and avoids Gaussian redundancy for improving the robustness and efficiency of inferring 3D Gaussians from sparse views. Extensive experiments on the LLFF, DTU, and Blender datasets demonstrate that our method significantly outperforms the state-of-the-art methods. Project page is available at: `https://hanl2010.github.io/Binocular3DGS/`.

## 1 Introduction

3D reconstruction technologies [28, 19] have demonstrated significant advances in synthesizing realistic novel views given a set of dense input views. To explore the challenging task in harsh real-world situations where only sparse inputs are available, some studies learn NeRF [28] with specially designed constraints [17, 44, 43, 49] and regularizations [30, 54, 50] on the view scarcity. However, NeRF-based methods often suffer from slow training and inference speeds, leading to high computational costs that restrict their practical applications.

3D Gaussian Splatting (3DGS) [19] has achieved notable advantages in rendering quality and efficiency. However, 3DGS is still facing severe challenges with inputting sparse views, where the unstructured 3D Gaussians with limited constraints tend to overfit the given few views, resulting in geometric inaccuracies for scene learning. Some recent studies [31, 71, 22, 51] on sparse view synthesis based on 3DGS employ the commonly-used depth priors from pre-trained models as additional constraints on the Gaussians geometries. However, the neural priors are often noisy and blurry, which struggle to precisely guide the learning of radiance fields.

In this paper, we aim to design a method that does not require external prior as supervision, which directly explores the self-supervisions from the few input views for improving the quality and

---

*The corresponding author is Yu-Shen Liu.

38th Conference on Neural Information Processing Systems (NeurIPS 2024).

efficiency of sparse 3DGS. We justify that the key factors in achieving this goal include 1) learning more accurate scene geometry of Gaussians which leads to consistent views synthesis, and 2) avoiding redundant Gaussians near the surface for better efficiency and filtering noises.

For learning more accurate scene geometry of Gaussians, we explore the self-supervisions inherent in the binocular stereo consistency to constrain on the rendered depth of 3DGS solely utilized from given input views and synthesized novel views. Our key insight lies in the observation that binocular image pairs implicitly involve the property of view consistency, as demonstrated in the binocular stereo vision methods [12, 11, 58]. Specifically, we first translate the camera of one input view slightly to the left or right to obtain a translational view, from which we render the image and depth from 3DGS. The rendered image and the input one thus form a left-right view pair as in binocular stereo vision. We then leverage the rendered depth and the known camera intrinsic to compute the disparity of the view pair. We conduct the supervisions by warping the rendered image of the translational view to the viewpoint of the input image using the disparity, and constrains on the consistency between the warped and input images.

To further reduce redundant Gaussians near the scene surface and enhance the quality and efficiency of novel view synthesis, we propose a decay schema for the opacity of Gaussians. Specifically, we simply apply a decay coefficient to the opacity property of the Gaussians, penalizing the opacity during training. To this end, Gaussians with lower opacity gradients (i.e., where the increase in opacity is smaller than the decay) are pruned, while Gaussians with higher opacity gradients (i.e., where the increase in opacity is greater than the decay) are retained. As the optimization process continues, redundant Gaussians are filtered out, and those newly generated (copied or split) Gaussians that are closer to the scene surface are retained, resulting in cleaner and more robust Gaussians. This opacity decay strategy significantly reduces artifacts in novel views and decreases the number of Gaussians, improving both the rendering quality and optimization efficiency of 3DGS.

Additionally, to achieve better geometry initialization for improving 3DGS quality when conducting optimization on sparse views, we use pre-trained keypoints matching network [42] to generate a dense initialization point cloud. The dense point cloud describes the geometry of the scene more accurately, preventing Gaussians from appearing far from the scene surface, especially in low-texture areas where the distribution of Gaussians is subject to limited constraints.

In summary, our main contributions are as follows.

- We propose a novel method for synthesizing novel views from sparse views with Gaussian Splatting that does not require external prior as supervision. We explore the self-supervisions inherent in the binocular stereo consistency to constrain the rendered depth, solely obtained from existing input views and synthesized views.

- We propose an opacity decay strategy which significantly regularizes the learning of Gaussians and reduces redundancy among Gaussians, leading to better rendering quality and optimization efficiency for novel view synthesis from sparse view with Gaussian Splatting.

- Extensive experiments on widely-used forward-facing and 360-degree scene datasets demonstrate that our method achieves state-of-the-art results compared to existing sparse novel view synthesis methods.

## 2 Related Works

### 2.1 Neural Radiance Field

Neural implicit functions have made great progress in surface reconstruction [45, 65, 16, 64, 56, 62, 63, 60], 3D representation [67, 53, 68, 23, 26, 66, 24] and generation [7, 70, 69, 47, 61, 25]. Detailed and realistic 3D scene representation has always been the research goal in the field of computer vision, and neural radiance fields (NeRFs) [28] has brought fundamental innovation to this domain. NeRF can reconstruct high-quality 3D scenes from sparse 2D images and generate realistic images from arbitrary viewpoints by representing scenes as continuous volume radiance functions.

However, NeRF requires a large number of views as input during the training stage and exhibits limitations in both training and inference speed. Consequently, the following researches have primarily focused on addressing these bottlenecks by improving computational efficiency [10, 29, 39,

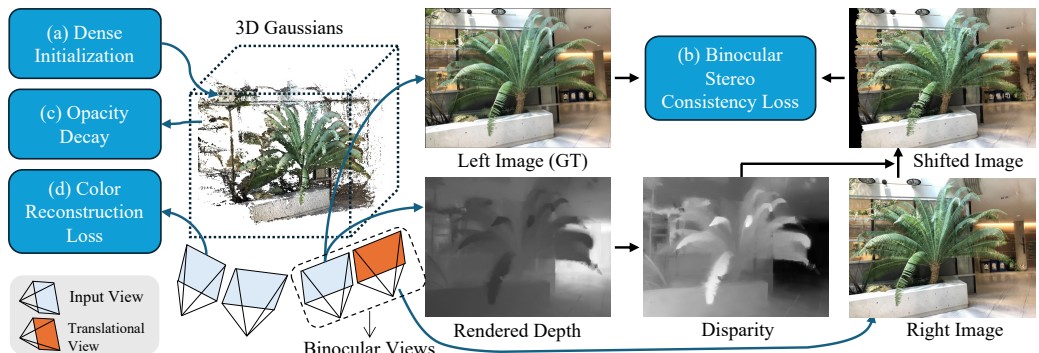

Figure 1: The overview of our method. (a) We leverage dense initialization for achieving Gaussian locations, and optimize the locations and Gaussian attributes with three constraints or strategies: (b) Binocular Stereo Consistency Loss. We construct a binocular view pair by translating an input view with camera positions, where we constrain on the view consistency of binocular view pairs in a self-supervised manner. (c) Opacity Decay Strategy is designed to decay the Gaussian opacity during training for regularizing them. (d) The Color Reconstruction Loss.

19, 15, 4, 5, 9] or reducing the number of input views [30, 17, 44, 43, 6, 49, 37, 21, 8, 38, 36, 40, 50], while continuously striving for enhanced rendering quality [1, 2, 3].

Notably, recent approach 3D Gaussian Splatting [19] has shown promising results in achieving real-time rendering capabilities without compromising rendering quality.

3D Gaussian Splatting [19] is an emerging method for novel view synthesis, which reconstructs high quality scenes rapidly by utilizing a set of 3D Gaussians to represent the radiance field in the scene. This method performs excellently when dealing with real-world scenes, particularly excelling in handling high-frequency details. Moreover, 3D Gaussian Splatting demonstrates significant advantages in inference speed and offers more intuitive editing and interpretability capabilities.

## 2.2 Novel View Synthesis from Sparse Views

Recent researches [30, 17, 44, 43, 6, 49, 37, 21, 8, 38, 36, 40, 50] have explored various approaches to generate novel views from sparse input images, focusing on enhancing both rendering quality and efficiency. Methods such as NeRF[28] and 3DGS [19] have been refined through various techniques, resulting in continuous improvement in the quality of novel view synthesis.

Using depth priors obtained from pre-trained networks [55, 32, 33] to supervise neural radiance fields is a widely used technique [38, 44]. Some methods [30, 54, 50, 8] introduce regularization terms in NeRF to address the problem, such as frequency regularization [54] and ray entropy regularization [20]. Additionally, there are also some methods [50, 49, 17] that leverage pre-trained models such as Diffusion model to enhance the rendering quality of novel views. However, most methods incur high costs during training and inference. While some methods have improved inference efficiency through generalizable models [5, 57, 52] or by using voxel grids [40], they often sacrifice rendering quality.

Currently, with the advent of 3D Gaussian splatting, some methods use 3DGS for sparse view synthesis, such as FSGS [71], SparseGS [51], DNGaussian [22] and CoherentGS [31]. These methods utilize depth priors obtained from pre-trained models [32] as supervision. Depth constraints enable the unstructured Gaussians to approximately distribute along the scene surface, thereby enhancing the quality of novel view images. However, the depth priors from pre-trained models often contain significant errors and cannot make the Gaussians distribute to optimal positions. In contrast, our method employs view consistency constraints based on binocular vision, resulting in more accurate depth information and thereby achieving a more optimal distribution of Gaussians. Some concurrent studies [48, 34] also employed the concept of binocular stereo in 3DGS [19], but they feed binocular images into a pre-trained network to obtain depth priors, instead of self-supervision.

# 3 Methods

The pipeline of our method is depicted in Figure 1. In this section, we first review the 3D representation method based on 3D Gaussians. Then, we explain how to construct stereo view pair and utilize it to enforce view consistency in a self-supervised manner. Next, we introduce the opacity decay strategy and the dense initialization method for 3D Gaussians. Finally, we present the overall loss function used for optimization.

## 3.1 Review of 3D Gaussian Splatting

Gaussian splatting [19] represents scene with a set of 3D Gaussians. Each 3D Gaussian is defined by a central location $\mu \in \mathbb{R}^3$, and a covariance matrix $\Sigma \in \mathbb{R}^{3 \times 3}$. Formally, it is defined as

$$G_i(x) = e^{-\frac{1}{2}(x-\mu_i)^T \Sigma^{-1}(x-\mu_i)}, \tag{1}$$

where the covariance matrix $\Sigma$ has physical meaning only when it is positive semi-definite. Therefore, it can be decomposed into $\Sigma = RSS^T R^T$, $S \in \mathbb{R}^3$ is a diagonal scaling matrix with 3 parameters, $R \in \mathbb{R}^4$ is a rotation matrix analytically expressed with quaternions. In addition, for rendering the image, each Gaussian also stores an opacity value $\alpha \in \mathbb{R}$ and a color feature $f \in \mathbb{R}^k$.

The 3D Gaussian is projected into the 2D image space when rendering an image, the projected 2D Gaussian is sorted by its depth value, and then alpha blending is used to calculate the color of each pixel,

$$c = \sum_{i=1}^{n} c_i \alpha_i' \prod_{j=1}^{i-1}(1 - \alpha_j'), \tag{2}$$

where $c_i$ is the color computed from feature $f$ and $\alpha'$ is the opacity of the 2D Gaussian, which is obtained by multiplying the covariance $\Sigma'$ of the 2D Gaussian by the opacity $\alpha$ of the corresponding 3D Gaussian. The 2D covariance matrix $\Sigma'$ is calculated by $\Sigma' = JW\Sigma W^T J^T$, where $J$ is the Jacobian of the affine approximation of the projective transformation. $W$ is the view transformation matrix.

The optimization process of Gaussian splatting is initialized with a set of 3D Gaussians from a sparse point cloud from SfM. Subsequently, the density of the Gaussian set is optimized and adaptively controlled. During optimization, a fast tile-based renderer is employed, allowing competitive training and inference times compared to the fast NeRF based methods.

## 3.2 Binocular Stereo Consistency Constraint

The key factor in improving the rendering quality of 3DGS is to guide the Gaussians to be distributed close to the exact scene surfaces. To this end, we aim to design a constraint on the rendered depth of 3DGS which directly represents the Gaussian geometry. Previous studies commonly adopt the depth priors from pretrained models to guide the depth of 3DGS. However, the depth priors are often noisy and blurry, especially for complex scenes, which fails in providing accurate guidance.

In this paper, we propose a novel prior-free method which leverages the binocular stereo consistency in 3D Gaussian splatting. We constrain on the rendered depth of 3DGS solely obtained by the existing input views and the novel views rendered by 3DGS. Specifically, we first render a novel view from a view point which is translated from one of the input views, leading to a pair of binocular vision images. We then shift the novel view to the perspective of the input image using disparity. Finally, we constrain the rendered image using the consistency between the input image and the warped image.

Specifically, given an image view $I_l$ as input, we translate its corresponding camera position $C_l$ to the right by a distance $d_{cam}$ and obtain the translational camera position $O_r$. We then obtain a novel rendered image $I_r$ by rendering 3D Gaussians from $O_r$. The images $I_l$ and $I_r$ form a binocular stereo image pair, where $I_l$ is the left image and $I_r$ is the right one, respectively. Simultaneously, we can obtain the rendered depth $D_l$ corresponding to the left image $I_l$ by rendering 3DGS from $O_l$. According to the geometric theory of binocular stereo vision, the relationship between disparity $d$ and depth $D_l$ is given by $d = f \cdot d_{cam}/D_l$, where $d \in \mathbb{R}^{h \times w}$ and each value $d_{ij}$ in $d$ indicates the horizontal shift required to align a pixel in the right image with its counterpart in the left image. $f$ is the focal length of the camera. Then, we can use the disparity $d$ to move each pixel of the right image

$I_r$, obtaining the reconstructed left image $I_{shifted}$, as denoted by

$$I_{shifted}[i,j] = I_r[i - d_i, j - d_j].$$ (3)

In practice, to make the reconstructed left image $I_{shifted}$ differentiable, we use a bilinear sampler to interpolate on the right image $I_r$ to obtain each pixel of $I_{shifted}$. The sampling coordinates for each pixel are directly obtained from the disparity. Formally, we define the loss function by

$$L_{consis} = \frac{1}{N} \sum_{i,j} \left| I_{ij}^l - I_{ij}^{shifted} \right|.$$ (4)

To achieve better optimization with the loss function, we do not use a fixed camera position shift in practice. Instead, we randomly translate the input camera to the left or right by a distance within a range $[-d_{max}, d_{max}]$ in each iteration.

Although some methods [38, 6, 21] use unseen views or neighboring training views as source views and warp them to the reference view for consistency constraints, they do not account for the impact of the distance and angle between the source and reference views on the effectiveness of the supervision. When the source view is rotated or moved away from the reference view, the warped image suffers severe distortion due to depth errors and occlusions. Compared to the ground truth (GT) image, this leads to larger errors, hindering the convergence of the image-warping loss. In contrast, slight camera translations that cause minor changes in view without rotation and negligible impact from occlusions, allow the errors between the warped image and the GT image to primarily stem from depth errors, facilitating better optimization of depth. For the results using different source views, please refer to the Appendix.

### 3.3 Opacity Decay Strategy

We further justify that relying solely on the depth constraints does not always lead to correct Gaussian geometries that are closely aligned with the exact scene surfaces. The reason is that the rendered depth varies with changes in the scale and opacity of the Gaussians, rather than being solely determined by their positions. While 3DGS flexibly optimizes the scale and opacity during training, which leads to deviations in the novel views. To address this issue, we design a simple strategy by applying a decay coefficient $\lambda$ to the opacity $\alpha$ of the Gaussians, penalizing the opacity during training.

$$\hat{\alpha} = \lambda\alpha, 0 < \lambda < 1.$$ (5)

We illustrate the opacity decay strategy in Figure 2. Assuming all Gaussians are in the initialized state, due to the accumulation of constraints from multiple views, Gaussians near the scene surface typically have larger opacity gradients, allowing opacity to rise rapidly and construct the scene surface. However, some Gaussians far from the surface fail to get their opacity decreased and be pruned due to insufficient multiview consistent constraints, eventually affecting the rendering quality of novel views.

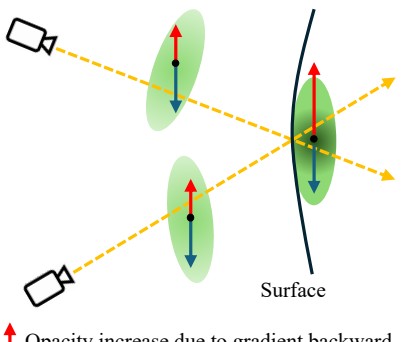

Opacity increase due to gradient backward

Opacity decay

Figure 2: Illustration of the Gaussian opacity decay strategy.

We aim to improve and stabilize the optimization of 3DGS by filtering out the far away Gaussians which indicates incorrect geometry while remain the ones close to the exact scene surfaces. This is achieved by applying the opacity decay strategy. We justify that the strategy does not lead all Gaussians' opacity going down, such as the ones on the surface, since these Gaussians' opacity progressively increase. As illustrated in Figure 2, the Gaussians far from the scene surface have lower opacity gradients due to fewer constraints, meaning their opacity increases less than those of the Gaussians on the scene surface. Under the opacity decay strategy, the opacity of Gaussians with lower opacity gradients gradually decreases until they are pruned. Conversely, the increase in opacity for Gaussians near the scene surface exceeds the decay magnitude, ultimately achieving a balance between the opacity increase and the decay, thereby preserving Gaussians close to the surface.

Table 1: Quantitative comparison on LLFF. We evaluate the NeRF-based and the 3DGS-based methods, our method achieves the best results in all metrics under different input-view settings.

| Methods | PSNR↑ | | | SSIM↑ | | | LPIPS↓ | | |
|---|---|---|---|---|---|---|---|---|---|
| | 3-view | 6-view | 9-view | 3-view | 6-view | 9-view | 3-view | 6-view | 9-view |
| DietNeRF [17] | 14.94 | 21.75 | 24.28 | 0.370 | 0.717 | 0.801 | 0.496 | 0.248 | 0.183 |
| RegNeRF [30] | 19.08 | 23.10 | 24.86 | 0.587 | 0.760 | 0.820 | 0.336 | 0.206 | 0.161 |
| FreeNeRF [54] | 19.63 | 23.73 | 25.13 | 0.612 | 0.779 | 0.827 | 0.308 | 0.195 | 0.160 |
| SparseNeRF [44] | 19.86 | 23.26 | 24.27 | 0.714 | 0.741 | 0.781 | 0.243 | 0.235 | 0.228 |
| ReconFusion [49] | 21.34 | 24.25 | 25.21 | 0.724 | 0.815 | 0.848 | 0.203 | 0.152 | 0.134 |
| MuRF [52] | 21.26 | 23.54 | 24.66 | 0.722 | 0.796 | 0.836 | 0.245 | 0.199 | 0.164 |
| 3DGS [19] | 15.52 | 19.45 | 21.13 | 0.405 | 0.627 | 0.715 | 0.408 | 0.268 | 0.214 |
| FSGS [71] | 20.31 | 24.20 | 25.32 | 0.652 | 0.811 | 0.856 | 0.288 | 0.173 | 0.136 |
| DNGaussian [22] | 19.12 | 22.18 | 23.17 | 0.591 | 0.755 | 0.788 | 0.294 | 0.198 | 0.180 |
| Ours | 21.44 | 24.87 | 26.17 | 0.751 | 0.845 | 0.877 | 0.168 | 0.106 | 0.090 |

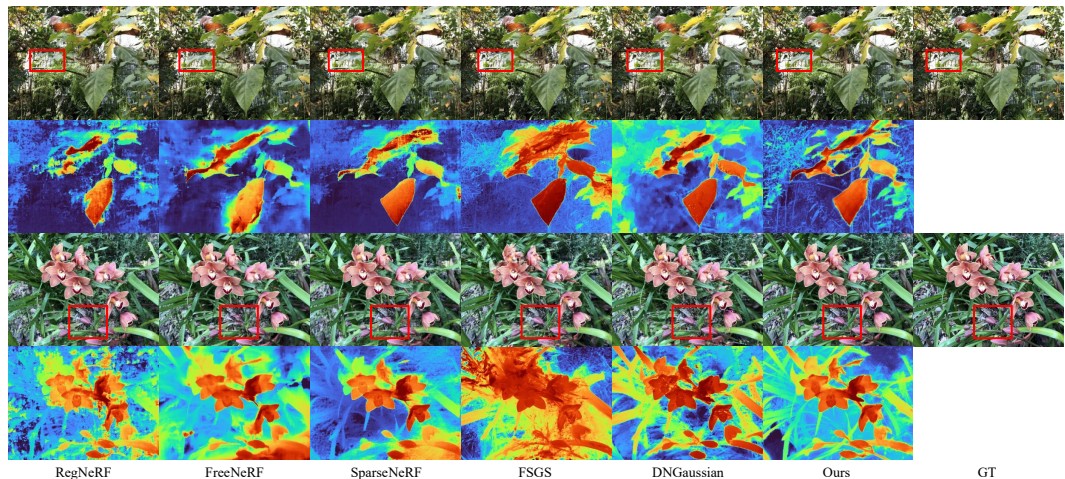

Figure 3: Visual comparison on LLFF dataset.

## 3.4 Initialization from dense point clouds

Previous 3DGS methods [19, 71, 51] usually utilize a sparse point cloud generated by Structure from Motion (SfM) [35] to initialize 3D Gaussians. However, the point cloud produced by sparse views is too sparse to adequately describe the scene to be reconstructed. Although the splitting strategy in 3DGS can replicate new Gaussians to cover the under-reconstructed area, they are subject to limited geometric constraints and cannot adhere well to the scene surfaces, especially for low-texture areas where the distribution of Gaussians may be arbitrary. We therefore seek a robust approach to achieve better geometry initialization for improving 3DGS quality when optimizing from sparse views. Note that this prior is just used for initialization, and we do not use any this kind of priors during learning 3D Gaussians.

To achieve this, we use a pre-trained keypoints matching network [42] to generate a dense initialization point cloud. Specifically, we arbitrarily select two images from the input images, input them into the matching network, and obtain matching points. We then leverage the triangulation method [14], along with the camera parameters corresponding to these images, to project the matching points into 3D space. This forms a dense point cloud, providing a more robust initialization for the Gaussians.

Compared with the sparse point cloud, the dense point cloud describes the geometry of the scene more accurately, preventing Gaussians from appearing far from the scene surface and ultimately leading to improved quality in novel view synthesis.

Table 2: Quantitative comparison on DTU. We evaluate the NeRF-based and the 3DGS-based methods, our method achieves the best results in most metrics under different input-view settings.

| Methods | PSNR↑ | | | SSIM↑ | | | LPIPS↓ | | |
|---|---|---|---|---|---|---|---|---|---|
| | 3-view | 6-view | 9-view | 3-view | 6-view | 9-view | 3-view | 6-view | 9-view |
| DietNeRF [17] | 11.85 | 20.63 | 23.83 | 0.633 | 0.778 | 0.823 | 0.214 | 0.201 | 0.173 |
| RegNeRF [30] | 18.89 | 22.20 | 24.93 | 0.745 | 0.841 | 0.884 | 0.190 | 0.117 | 0.089 |
| FreeNeRF [54] | 19.52 | 23.25 | 25.38 | 0.787 | 0.844 | 0.888 | 0.173 | 0.131 | 0.102 |
| SparseNeRF [44] | 19.47 | - | - | 0.829 | - | - | 0.183 | - | - |
| ReconFusion [49] | 20.74 | 23.62 | 24.62 | 0.875 | 0.904 | 0.921 | 0.124 | 0.105 | 0.094 |
| MuRF [52] | 21.31 | 23.74 | 25.28 | 0.885 | 0.921 | 0.936 | 0.127 | 0.095 | 0.084 |
| 3DGS [19] | 10.99 | 20.33 | 22.90 | 0.585 | 0.776 | 0.816 | 0.313 | 0.223 | 0.173 |
| FSGS [71] | 17.34 | 21.55 | 24.33 | 0.818 | 0.880 | 0.911 | 0.169 | 0.127 | 0.106 |
| DNGaussian [22] | 18.91 | 22.10 | 23.94 | 0.790 | 0.851 | 0.887 | 0.176 | 0.148 | 0.131 |
| Ours | 20.71 | 24.31 | 26.70 | 0.862 | 0.917 | 0.947 | 0.111 | 0.073 | 0.052 |

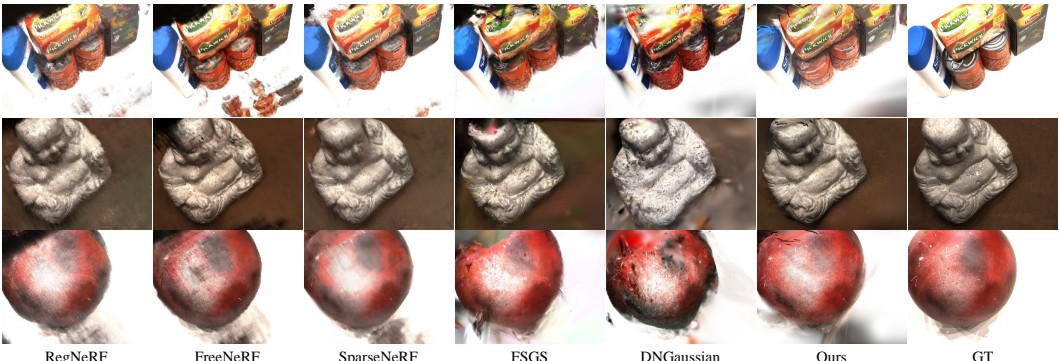

| RegNeRF | FreeNeRF | SparseNeRF | FSGS | DNGaussian | Ours | GT |

Figure 4: Visual comparison on DTU dataset.

## 3.5 Training Loss

The final loss function consists of two parts: the proposed binocular stereo consistency loss $L_{consis}$ as introduced in Eq. (4) and the commonly-used color reconstruction loss $L_{color}$ of 3DGS [19]. We define the overall loss function by

$$L = L_{consis} + L_{color}, \tag{6}$$

where $L_{color}$ is composed of an $L_1$ loss and a structural similarity loss $L_{D-SSIM}$, as denoted by

$$L_{color} = (1 - \beta)L_1 + \beta L_{D-SSIM}. \tag{7}$$

## 4 Experiments

### 4.1 Datasets

We conduct experiments on three public datasets, including the LLFF dataset [27], the DTU dataset [18] and the NeRF Blender Synthetic dataset (Blender) [28]. Following prior works [30, 54, 17], we used 3, 6, and 9 views as training sets for the LLFF and DTU datasets, and 8 images for training on the Blender dataset. The selection of test images remained consistent with previous works [17, 30, 54]. The downsampling rates for the LLFF, DTU, and Blender datasets are 8, 4, and 2, respectively.

### 4.2 Implementation details

Since the LLFF and DTU are datasets of forward-facing scenes, they cannot be constrained by views from other directions during the optimization process. On the other hand, Blender is a dataset of 360-degree scenes, 8 images from different viewpoints are used as input during training, providing

stronger constraints. Therefore, for the LLFF and DTU datasets, we utilize a pre-trained matching network PDC-Net+ [42] to obtain keypoints from input images, which are then used as the dense initialization point clouds. For the Blender dataset, we adopt random initialization as in the original 3DGS [19].

Moreover, based on the analysis above, we train the LLFF and DTU datasets for 30,000 iterations, while the Blender dataset is trained for 7,000 iterations. Since the view consistency constraint based on binocular stereo vision needs to be performed on the basis of the training views that can already be rendered with high quality, we add the view consistency loss at 20,000 iterations for the LLFF and DTU datasets, and at 4,000 iterations for the Blender dataset. The maximum distance $d_{max}$ for camera shift is set to 0.4, the opacity decay coefficient $\lambda$ is set to 0.995, and the $\beta$ in the loss function 7 is set to 0.2 as in the original 3DGS [19].

Table 3: Quantitative comparison on Blender for 8 input views. We evaluate the NeRF-based and the 3DGS-based methods, our method achieves comparable results to the state of the art methods.

| Methods | PSNR↑ | SSIM↑ | LPIPS↓ |
|---|---|---|---|
| DietNeRF [17] | 22.50 | 0.823 | 0.124 |
| RegNeRF [30] | 23.86 | 0.852 | 0.105 |
| FreeNeRF [54] | 24.26 | 0.883 | 0.098 |
| SparseNeRF [44] | 22.41 | 0.861 | 0.199 |
| 3DGS [19] | 22.23 | 0.858 | 0.114 |
| FSGS [71] | 22.76 | 0.829 | 0.157 |
| DNGaussian [22] | 24.31 | 0.886 | 0.088 |
| Ours | 24.71 | 0.872 | 0.101 |

### 4.3 Baseline

We choose some state-of-the-art NeRF-based and 3DGS-based sparse view synthesis methods for comparison. NeRF-based methods include DietNeRF [17], RegNeRF [30], FreeNeRF [54], SparseNeRF [44], ReconFusion [49] and MuRF [52]. 3DGS-based methods include DNGaussian [22] and FSGS [71]. Additionally, we compare against the original 3DGS [19].

### 4.4 Comparisons

**LLFF.** Table 1 shows the quantitative results of the LLFF dataset with 3, 6, and 9 input views, respectively. Our method surpasses all baseline methods in terms of PSNR, SSIM [46], and LPIPS [59] scores under different number of input views. When the number of input views increases to 9, it is almost adequate to provide sufficient color constraints. However, the evaluation scores of DNGaussian [22] do not show a significant improvement, indicating that it does not perform well with an increased number of input views. This is because errors in the depth prior negatively affect the optimization process.

Figure 3 presents the visual comparison of novel view synthesis and depth rendering for the *leaves* and *orchids* scenes in the LLFF dataset. From the rendered results of novel views, several state-of-the-art methods are all perceptually acceptable. However, regarding the depth maps, RegNerf [30], FreeNerf [54], and FSGS [71] exhibit poor quality, indicating significant errors in their geometric estimation. SparseNeRF [44] and DNGaussian [22] indirectly utilize relative depth information from depth priors to supervise geometry estimation, hence achieving adequate rendering depth. Although FSGS [71] also exploits depth priors, it attempts to directly employ depth prior information by using depth correlation loss. It does not yield accurate scene geometry,

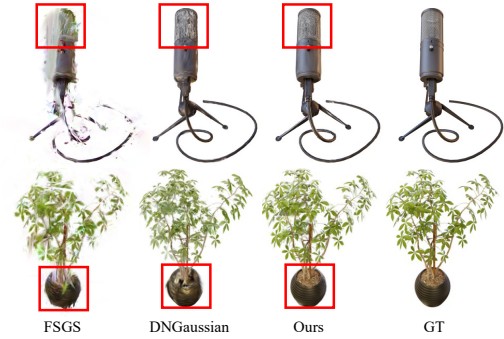

FSGS  DNGaussian  Ours  GT

Figure 5: Visual comparison on Blender dataset.

instead, inaccurate depth prior information exacerbates scene geometry. In contrast, our method utilizes self-supervised view consistency loss to obtain more precise geometry, thereby recovering more structural details in novel views.

**DTU.** Table 2 shows the quantitative results of the DTU dataset under 3, 6, and 9 input views respectively. MuRF [52] performs better on the DTU dataset when using 3 views as input. However, generalizable methods require large scale datasets and time-consuming training to learn a prior. Figure 4 illustrates the visual comparison of novel view synthesis for three scenes in the DTU dataset.

Table 4: Ablation studies on LLFF and DTU dataset with 3 input views.

| Dense Init | $L_{consis}$ | Opacity Decay | LLFF | | | DTU | | |
|---|---|---|---|---|---|---|---|---|
| | | | PSNR↑ | SSIM↑ | LPIPS↓ | PSNR↑ | SSIM↑ | LPIPS↓ |
| | | | 15.52 | 0.405 | 0.408 | 10.99 | 0.585 | 0.313 |
| ✓ | | | 19.30 | 0.651 | 0.238 | 16.36 | 0.778 | 0.181 |
| | ✓ | | 17.53 | 0.522 | 0.322 | 13.41 | 0.773 | 0.202 |
| | | ✓ | 18.95 | 0.606 | 0.268 | 14.40 | 0.723 | 0.219 |
| ✓ | ✓ | | 20.48 | 0.715 | 0.218 | 19.08 | 0.832 | 0.131 |
| | ✓ | ✓ | 20.82 | 0.716 | 0.189 | 15.33 | 0.739 | 0.207 |
| ✓ | | ✓ | 21.13 | 0.738 | 0.180 | 17.12 | 0.812 | 0.152 |
| ✓ | ✓ | ✓ | 21.44 | 0.751 | 0.168 | 20.71 | 0.862 | 0.111 |

To distinctly discern the differences among various methods, we select the synthesized images far from the training views for comparison in each scene. From the rendered results of novel views, it can be observed that NeRF-based methods produce blurry results, while FSGS [71] and DNGaussian [22] exhibit numerous artifacts due to insufficient constraints on Gaussians. Our method outperforms previous state-of-the-art methods on most evaluation metrics and achieves better visual quality as well.

**Blender.** We evaluate our method on 360-degree scenes using the Blender dataset. Table 3 presents the quantitative results under 8 input views. Our approach outperforms all baselines in the PSNR score, and achieves comparable SSIM and LPIPS. We could not reproduce the results reported in the FSGS [71] on the Blender dataset and used the own test results in Table 3. Figure 5 illustrates the visual comparison results of several 3DGS-based methods. It can be observed that FSGS [71] exhibits noticeable artifacts in the synthesized images due to insufficient Gaussian constraints, and the rendering results of DNGaussian [22] also has some distortions. In contrast, our method performs better in terms of detail preservation.

## 4.5 Ablation Study

To verify the effectiveness of the view consistency loss, opacity decay strategy and the dense initialization for Gaussians, we isolate each of these modules separately while keeping the other modules unchanged. We then evaluate the metrics and illustrate the visual results. As shown in Tables 4 and 5, the performance decreases when removing any of the modules we proposed.

Table 5: Ablation studies on Blender dataset with 3 input views.

| $L_{consis}$ | Opacity Decay | PSNR ↑ | SSIM ↑ | LPIPS ↓ |
|---|---|---|---|---|
| | | 22.23 | 0.858 | 0.114 |
| ✓ | | 23.47 | 0.861 | 0.112 |
| | ✓ | 23.96 | 0.865 | 0.109 |
| ✓ | ✓ | 24.71 | 0.872 | 0.101 |

**Effectiveness of view consistency loss.** To verify the effectiveness of the view consistency loss, we compare the depth maps from novel views. Figure 6 shows the visual comparison of rendered depth maps. The comparison in the figure shows that it is evident that the view consistency loss significantly improves the estimation of scene geometry.

**Effectiveness of opacity decay.** Figure 7 (b) and (c) show the comparison of novel view synthesis results and the visualization of Gaussian centers for the *trex* scene in LLFF before and after applying the opacity decay. It can be seen that without the opacity decay, Gaussians appear far from the surfaces and there is noise near the surfaces. After applying the opacity decay strategy, the Gaussians far from the scene surfaces are eliminated, and the noisy point clouds are significantly reduced, thereby further improving the rendering quality of novel views.

**Effectiveness of dense initialization.** Figure 7 (a) and (c) show the novel view images and the final Gaussian point clouds obtained using different initialization point cloud. It can be seen that without dense point cloud initialization, the spatial positions of the Gaussians become arbitrary in some low-texture regions or areas occluded in the training views due to insufficient constraints, resulting in reduced quality of the novel view images.

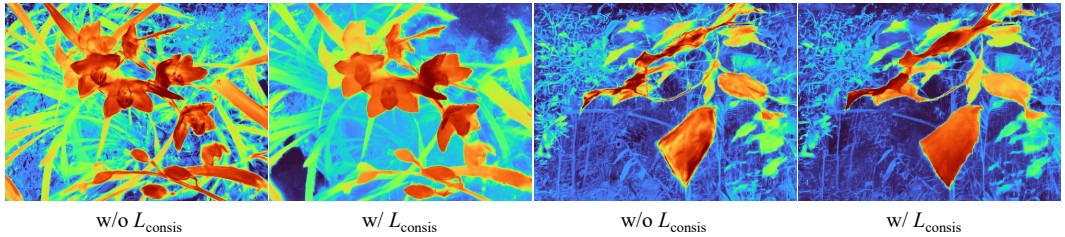

| w/o $L_{consis}$ | w/ $L_{consis}$ | w/o $L_{consis}$ | w/ $L_{consis}$ |

Figure 6: Visual comparison of depth maps before and after using view consistency loss.

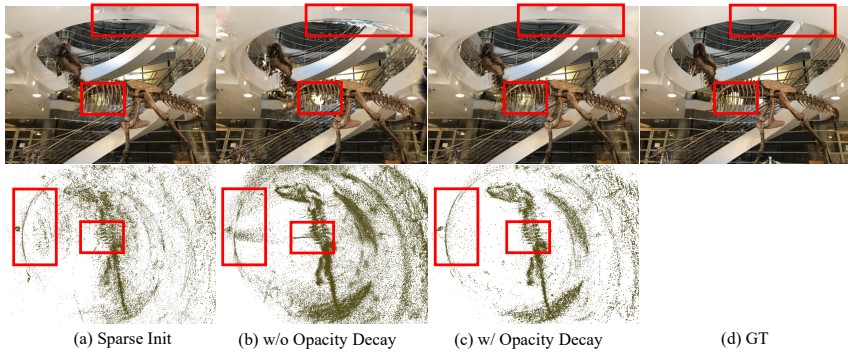

| (a) Sparse Init | (b) w/o Opacity Decay | (c) w/ Opacity Decay | (d) GT |

Figure 7: Visual comparison of novel view images and Gaussian point clouds.

## 5 Conclusion

In this paper, we propose a novel method for novel view synthesis from sparse views with 3DGS. We construct a self-supervised multi-view consistency constraint using the rendered and input images, and introduce a Gaussian opacity decay and a dense point cloud initialization strategy. These constraints ensure that the Gaussians are distributed as closely as possible to the scene surfaces and filter out those far from the surfaces. Our approach enables the unstructured Gaussians to accurately represent scene geometry even with sparse input views, resulting in high-quality novel rendering images. Extensive experiments on the LLFF, DTU and Blender datasets demonstrate that our method outperforms existing state-of-the-art sparse view synthesis methods.

**Limitation.** Since our method utilizes the view consistency constraints for estimating scene depth, some scene areas with low texture may lead to the inaccurate depth estimation (e.g. the white background areas in the DTU dataset), thus failing to constrain the corresponding Gaussians. This results in the white Gaussians potentially occluding the object in the novel views. In contrast, DNGaussian [22] uses the depth prior estimated by the pre-trained network to constrain the Gaussians, preventing this scenario from happening. The visual comparisons are presented in the Appendix.

## 6 Acknowledgement

This work was supported by National Key R&D Program of China (2022YFC3800600), the National Natural Science Foundation of China (62272263, 62072268), and in part by Tsinghua-Kuaishou Institute of Future Media Data.

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

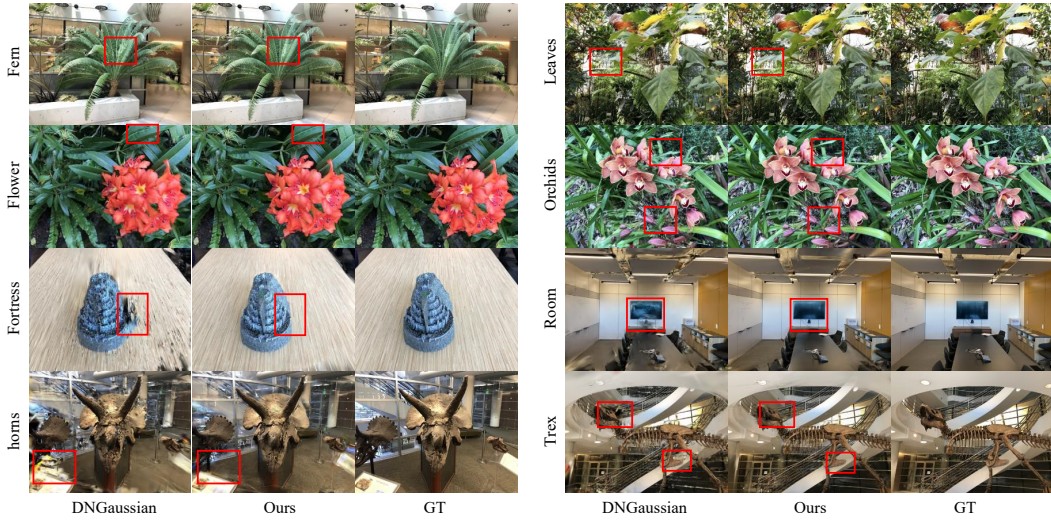

Figure A: Visual comparison on LLFF dataset with 3 input views.

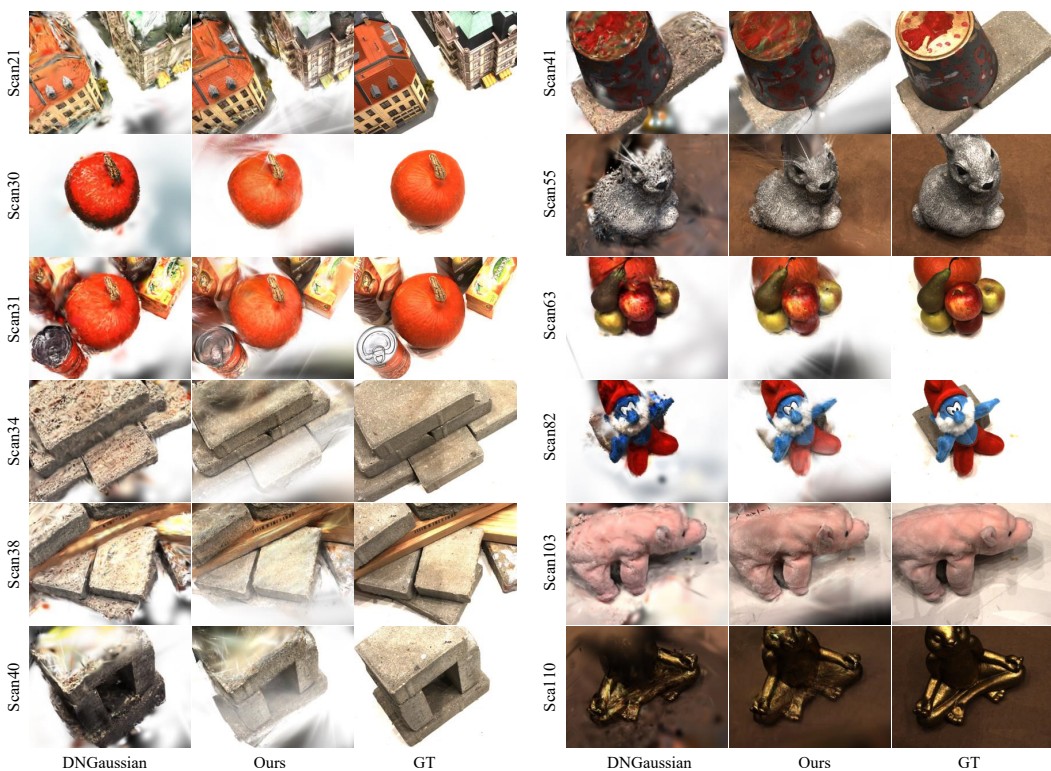

Figure B: Visual comparison on DTU dataset with 3 input views.

# A  More Visualizations

## A.1  Additional Results on LLFF

Figure A shows the visualization comparison of our method and DNGaussian [22] across all scenes in the LLFF [27] dataset. Each scene is trained using the same 3 input views. Although both methods achieve perceptually acceptable results, the novel view rendering results of our method are closer to the Ground Truth.

| Components | PSNR↑ | SSIM↑ | LPIPS↓ |
|---|---|---|---|
| Opacity Entropy Reg. | 15.75 | 0.480 | 0.361 |
| Opacity Decay | 21.44 | 0.751 | 0.168 |

Table A: Quantitative comparison of using different regularization components for opacity.

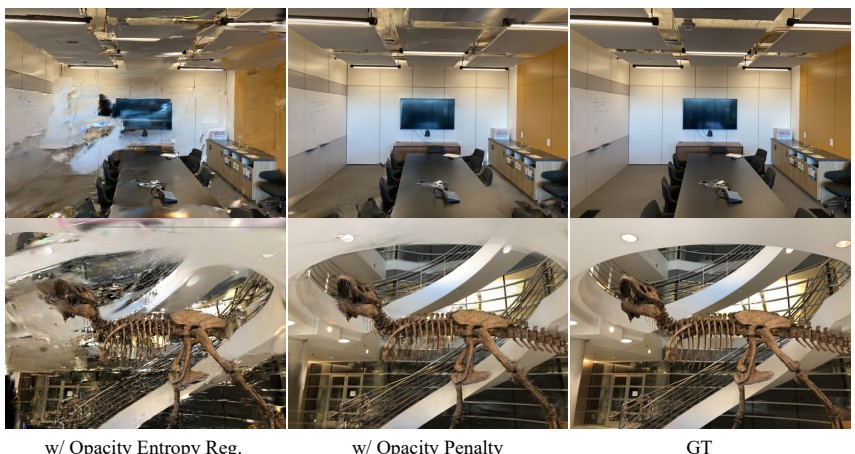

  w/ Opacity Entropy Reg.    w/ Opacity Penalty      GT

Figure C: Visual comparison of using different regularization components for opacity.

## A.2 Additional Results on DTU

Figure B shows an additional visual comparison between our method and DNGaussian [22] on the DTU [18] dataset. Each scene is trained using the same 3 input views. To clearly distinguish the differences between the two methods, we select a novel unseen view that is far from the training views for comparison. As analyzed in the limitations section of the main text, our method cannot effectively constrain the white background in training views, resulting in objects being occluded by white Gaussians in some scenes. Even so, the rendering quality of the novel views from our method is still better than the results from DNGaussian.

## B More Experiments

### B.1 Entropy regularization instead of Opacity Decay

The Sugar [13] method applies entropy regularization to the opacity of Gaussians to bring Gaussians closer to the surface, similar to our proposed opacity decay strategy. Therefore, we also evaluate the performance of opacity entropy regularization on the LLFF dataset. Following Sugar [13], the opacity threshold for Gaussians pruning is set to 0.5. Table A shows the metrics with opacity entropy regularization. It can be seen that our opacity decay strategy has advantages. Figure C illustrates the novel view synthesis using different opacity regularization strategies. It can be seen that entropy regularization leads to noticeable overfitting with sparse view inputs, resulting in lower quality novel view rendering.

### B.2 Training the DTU with Mask

To eliminate the undesirable effects of the solid color background on novel view rendering results in the DTU dataset, we perform additional experiments by using masks to filter out the background in the training views. The evaluation results are shown in Table B. Figure D illustrates the visual comparison between our method and DNGaussian [22] when using masks in the training views. It can be seen that without the solid color background, our method shows a significant improvement, while the performance of DNGaussian decreases.

| Methods | PSNR↑ | SSIM↑ | LPIPS↓ |
|---|---|---|---|
| DNGaussian [22] | 18.91 | 0.790 | 0.176 |
| DNGaussian$_{mask}$ [22] | 14.94 | 0.699 | 0.237 |
| Ours | 20.71 | 0.862 | 0.111 |
| Ours$_{mask}$ | 22.03 | 0.875 | 0.098 |

Table B: Quantitative comparison of using background masks for input views on DTU dataset.

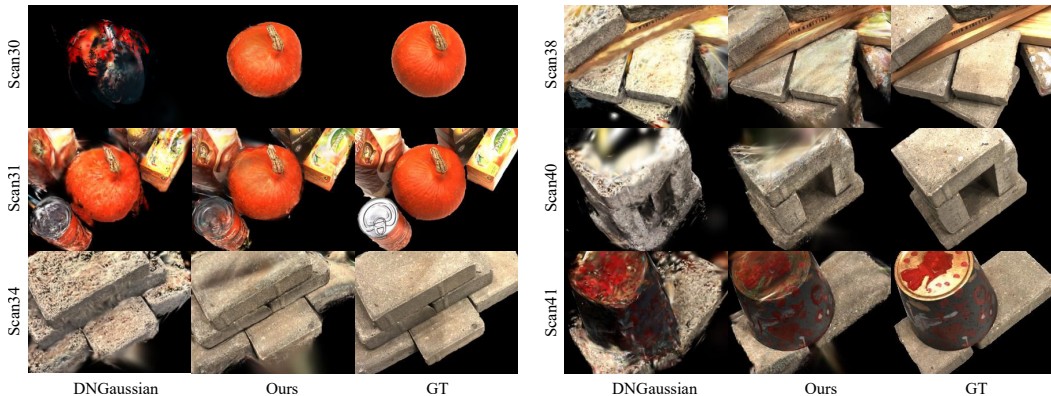

Figure D: Visual comparison of using background masks for input views on DTU dataset.

## C  Ablation studies for hyperparameters

### C.1  Ablation studies for hyperparameter $\lambda$

We perform ablation studies on the LLFF [27] and DTU [18] datasets with the value of $\lambda$ ranging from 0.96 to 1.0, and the results are shown in Table C. We find that the best performance is achieved when the value of $\lambda$ is 0.995 on both datasets. Therefore we set $\lambda$ to 0.995 for all datasets.

Table C: Ablation studies for $\lambda$ on LLFF dataset with 3 input views.

| $\lambda$ | | 0.960 | 0.970 | 0.975 | 0.980 | 0.985 | 0.990 | 0.995 | 0.998 | 1.0 |
|---|---|---|---|---|---|---|---|---|---|---|
| | PSNR↑ | 18.47 | 20.72 | 20.90 | 21.14 | 21.39 | 21.42 | 21.44 | 21.27 | 20.21 |
| LLFF | SSIM↑ | 0.629 | 0.711 | 0.712 | 0.730 | 0.742 | 0.748 | 0.751 | 0.745 | 0.709 |
| | LPIPS↓ | 0.344 | 0.237 | 0.223 | 0.202 | 0.184 | 0.172 | 0.168 | 0.171 | 0.201 |
| | PSNR↑ | 17.06 | 18.11 | 19.60 | 19.63 | 19.77 | 20.49 | 20.71 | 20.67 | 19.08 |
| DTU | SSIM↑ | 0.785 | 0.813 | 0.846 | 0.848 | 0.853 | 0.861 | 0.862 | 0.862 | 0.832 |
| | LPIPS↓ | 0.223 | 0.196 | 0.158 | 0.152 | 0.139 | 0.121 | 0.111 | 0.109 | 0.131 |

### C.2  Ablation studies for hyperparameter $d_{max}$

We perform ablation studies for $d_{max}$ on the LLFF[27] and DTU [18] datasets. The value of $d_{max}$ gradually increases from 0.1 to 0.8, we obtain the results as shown in Table D. We find that there is almost no performance increase on both the LLFF and DTU datasets when $d_{max}$ is greater than 0.4, therefore we set the $d_{max}$ to 0.4 for all datasets.

## D  The impact of initialization on overall performance

We conduct experiments on the LLFF [27] and DTU [18] datasets with several different initialization, including random initialization, sparse initialization, and using the LoFTR [41] matching network to construct initial point clouds. The results are shown in Table E.

Table D: Ablation studies for $d_{max}$ on LLFF and DTU dataset with 3 input views

| $d_{max}$ | LLFF | | | DTU | | |
|---|---|---|---|---|---|---|
| | PSNR↑ | SSIM↑ | LPIPS↓ | PSNR↑ | SSIM↑ | LPIPS↓ |
| 0.1 | 21.32 | 0.742 | 0.176 | 19.47 | 0.844 | 0.127 |
| 0.2 | 21.41 | 0.746 | 0.172 | 19.99 | 0.852 | 0.120 |
| 0.4 | 21.44 | 0.751 | 0.168 | 20.71 | 0.862 | 0.111 |
| 0.6 | 21.43 | 0.748 | 0.168 | 20.65 | 0.863 | 0.111 |
| 0.8 | 21.44 | 0.750 | 0.169 | 20.66 | 0.864 | 0.110 |

Table E: Impact of different initializations on performance.

| Init type | LLFF | | | DTU | | |
|---|---|---|---|---|---|---|
| | PSNR↑ | SSIM↑ | LPIPS↓ | PSNR↑ | SSIM↑ | LPIPS↓ |
| random init | 17.69 | 0.548 | 0.302 | 16.90 | 0.769 | 0.179 |
| sparse init | 20.82 | 0.716 | 0.189 | 15.33 | 0.739 | 0.207 |
| points by LoFTR [41] | 20.33 | 0.686 | 0.211 | 18.97 | 0.834 | 0.132 |
| points by PDCNet+ [42] | **21.44** | **0.751** | **0.168** | **20.71** | **0.862** | **0.111** |

PDCNet+ [42] can obtain more matching points than LoFTR [41], so the initialized point cloud obtained by PDCNet+ results in better performance. For the LLFF [27] dataset, the input views have large overlap, SfM methods can generate point clouds with better quality than LoFTR. Therefore, using sparse point clouds generated by SfM as initialization yields better performance than LoFTR. However, for the DTU [18] dataset, where there is small overlap among input views, SfM-generated point clouds are too sparse. Thus, the performance is even worse than random initialization.

## E   The impact of different source views on overall performance

Figure E shows the warped images and error maps when using different source views. It can be observed that when the source view is rotated relative to the reference view or is far away from the reference view, the warped image suffers from significant distortions due to depth errors and occlusions. This results in a large error compared to the GT image.

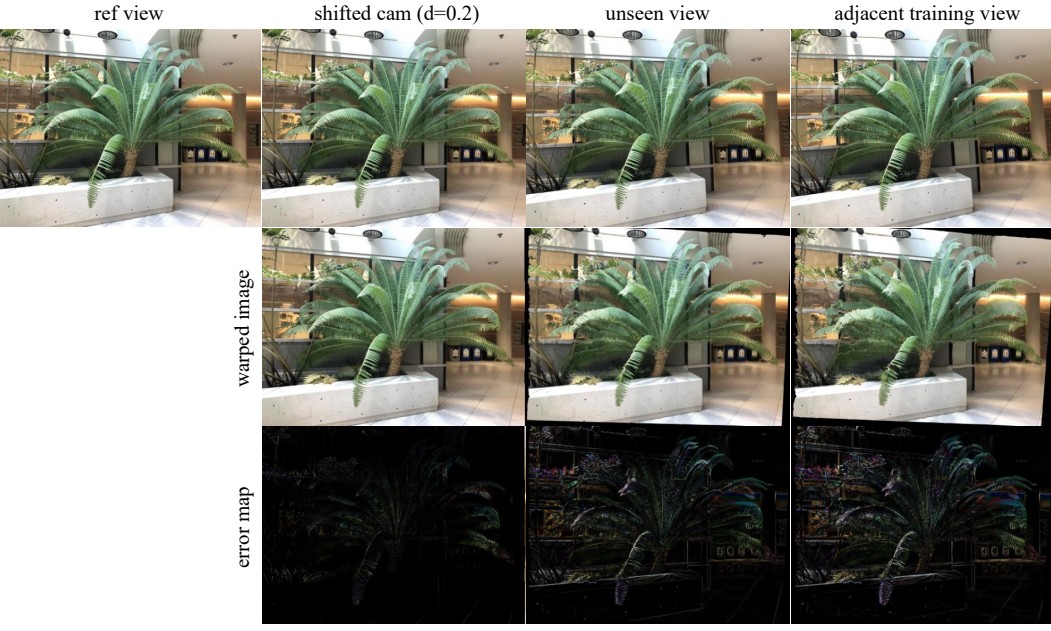

Figure E: The warped images and error maps when using different source views.

Table F shows the quantitative comparison using different source images on the LLFF and DTU datasets. The binocular vision-based view consistency method achieves the best performance.

Table F: Quantitative comparison using different source images on the LLFF and the DTU dataset.

| source view type | LLFF dataset | | | DTU dataset | | |
|---|---|---|---|---|---|---|
| | PSNR↑ | SSIM↑ | LPIPS↓ | PSNR↑ | SSIM↑ | LPIPS↓ |
| adjacent training view | 20.61 | 0.711 | 0.204 | 19.55 | 0.833 | 0.137 |
| unseen view | 20.88 | 0.728 | 0.185 | 19.96 | 0.850 | 0.124 |
| shifted cam | **21.44** | **0.751** | **0.168** | **20.71** | **0.862** | **0.111** |

# F   Implementations Details

Since our method uses an opacity decay strategy, we do not use the opacity reset strategy from the original 3DGS, and we also remove the scale threshold for Gaussians pruning from the original 3DGS.

We train our model on an RTX 3090 GPU, running 30,000 iterations per scene on the LLFF and DTU datasets, which takes approximately 10 minutes each. For the Blender dataset, we train for 7,000 iterations per scene, which takes approximately 3 minutes each. The storage space required for the Gaussian point clouds is about one-third of that required by the original 3DGS.

# G   Dataset Details

**LLFF.**   The LLFF [27] is a forward-facing dataset with 8 scenes. Following previous works [30, 54], we take every 8th image as the novel views for evaluation. The input views are evenly sampled from the remaining views. During training and evaluation, the images are downsampled by a factor of 8, resulting in a resolution of 378×504 for each image.

**DTU.**   The DTU [18] dataset consists of 124 scenes. We follow previous works [30, 54] to train and evaluate our method on 15 test scenes. The test scene IDs are: 8, 21, 30, 31, 34, 38, 40, 41, 45, 55, 63, 82, 103, 110, and 114. In each scene, we use images with the following IDs as input views: 25, 22, 28, 40, 44, 48, 0, 8, 13. The first 3 and 6 image IDs correspond to the input views in 3-view and 6-view settings respectively. We use 25 images as novel views for evaluation, with the following IDs: 1, 2, 9, 10, 11, 12, 14, 15, 23, 24, 26, 27, 29, 30, 31, 32, 33, 34, 35, 41, 42, 43, 45, 46, 47. During training and evaluation, the images are downsampled by a factor of 4, resulting in a resolution of 300 × 400 for each image.

**Blender.**   The Blender [28] dataset consists of 8 synthetic scenes. For fair comparison, we follow previous works [30, 54] and use 8 images as input views for each scene, with the following IDs: 26, 86, 2, 55, 75, 93, 16, 73, 8. The 25 test views are sampled evenly from the testing images for evaluation. During training and evaluation, the images are downsampled by a factor of 2, resulting in a resolution of 400 × 400 for each image.

