# OpenReview forum: "Binocular-Guided 3D Gaussian Splatting with View Consistency for Sparse View Synthesis"
_NeurIPS.cc/2024/Conference — NeurIPS 2024 poster_

### Official Review · Reviewer_gh5j · 2024-06-12

**Soundness:** 2
**Presentation:** 3
**Contribution:** 2
**Rating:** 4
**Confidence:** 4

**Summary:**

This paper proposed a 3D Gaussian Splatting method to render novel views with sparse inputs. This paper first use a pre-trained keypoints matching network to generate dense point initializations, and propose a consistent loss between the warped binocular stereo image. Meanwhile, they introduce a opacity penalty strategy to further learn the accurate gaussian points. The proposed method achieves the state-of-the-art performance on LLFF, DTU and Blender datasets.

**Strengths:**

1. The dense point generated from the pre-trained keypoints matching network can provide much better initialization for 3DGS with sparse inputs.
2. Using the generated binocular views to add the consistent loss can introduce more constraints in the sparse scenario.
3. The reported results seems to be much better than existing methods.

**Weaknesses:**

1. While the reported results show that the opacity penalty strategy brings the best improvement, it's motivation is not clear.
- Why add a coefficient to the opacity can guide the gaussian point to be closer to the scene surface? (L52. "guiding the remaining Gaussians to be closer to the scene surface"), especially under the setting of $\lambda=0.995$ (has a minor difference with 1.0), and are there any ablation results with different value configurations of $\lambda$. Because the coefficient only works on the opacity, the opinion of guiding the position of Gaussian to be closer to the scene surface does not convince me enough.
- The statement of "the gaussian close to surface has larger gradient and the gaussian far from the surface has smaller gradient" (Line176-179) seems unconvincing. The accurate Gaussian near the surface can render accurate colors with relatively small computational loss and should have small gradients. And on the contrary, the wrong Gaussian may has a larger gradient, like the situation of "wall" in the overfitting scenario.
- Such a small modification can bring nearly 2dB improvement is shocking, and the explanation and content of this part is not adequate. Adding more analysis might help make it more convincing.
2. The proposed binocular stereo consistency is actually a method to regularize the model using unseen virtual views.
- And using the virtual unseen view is a demonstrated existing strategy that has been adapted by many methods like [1,2,3], and the difference of this method is using the different sampling space of virtual views (the sampling space in this paper only has horizontal translation and no rotation).
- Intuitively, using more diverse sampling space like [1,2,3] may be more powerful, because it can use more unseen views from different angles to regularize the model. So what are the advantages of sampling only in the horizontal direction and are there any experimental comparisons?
- The disparity-based or depth-based warping will introduce the occlusion, which has a great impact of the performance. And how to mitigate this is not clarified in this paper.
- There might be an inaccuracy about the method. It seems like you want to use the backward warping to warp the right image to the left camera coordinate (Line148-160). Thus the depth needs to be pixel-aligned with the left image in Eq. (3), but you just use the depth of right image. This just confuses me a lot.
3. It seems that the dense initialization using the keypoints matching network plays an important role in the overall performance.
- How about the sensitivity of the method to different matching network?
- How about the performance of the method w/o dense or sparse initialization and only w/ the random initialization like DNGaussian. Because in the sparse scenario, popular SFM methods are hard to work and always fail to generate accurate points.
4. There are also some confusions in the experiment.
- The results reported by FSGS differ significantly from those in the paper, and there seems to be no explanation for this in the paper.
- Some visualization results (e.g., Fig. 3, Fig. 4 and Fig. 9) are not labeled with the corresponding methods, which makes comparison difficult.
- Since the runtime of  3DGS-based method is relatively fast, reporting the error bar or standard deviations of multiple experiments can help to enhance the credibility of the paper.

[1] SPARF: Neural Radiance Fields from Sparse and Noisy Poses, CVPR2023.

[2] GeCoNeRF: Few-Shot Neural Radiance Fields via Geometric Consistency, ICML2023.

[3] RegNeRF: Regularizing Neural Radiance Fields for View Synthesis from Sparse Inputs, CVPR2022.

**Questions:**

Refer to weaknesses for details. And due to these doubts, I tend to give the borderline and hope to see the author's response.

**Limitations:**

Have declared in the paper.

---

> ### Author Rebuttal · Authors · 2024-08-07
>
> We sincerely appreciate the reviewer gh5j for the invaluable feedback and time invested in evaluating our work. We respond to each question below.
>
> **_Q1: The opacity penalty guides the remaining Gaussians closer to the scene surface._**
>
> Opacity penalty prune the Gaussians that far from the scene surface as much as possible, while preserving the Gaussians close to the scene surface. Our description may be ambiguous here, and we will revise it in the revision.
>
> We perform ablation studies on the LLFF dataset with the value of $\lambda$ ranging from 0.96 to 1.0, and the results are shown in the **Table G**. We find that the best performance is achieved when the value of $\lambda$ is 0.995.
>
> **_Q2: "the Gaussian close to surface has larger gradient and the Gaussian far from the surface has smaller gradient"._**
>
> Our description may not be clear enough, the "gradient" refers to opacity. For instance, when initialized with a random point cloud, during the optimization process, Gaussians that are far from the scene surface will have their opacity reduced, thereby being pruned. Gaussians closer to the scene surface will see an increase in opacity, thus being preserved. However, not all Gaussians close to the surface have precise positions, hence, we employ opacity regularization to further eliminate Gaussians deviating from the surface, thereby enhancing the quality of novel view images.
>
> **_Q3: Explanation and content of Opacity Penalty Strategy is not adequate, adding more analysis._**
>
> We provide more detailed ablation studies on the LLFF, DTU, and Blender datasets, as shown in **Table B** and **Table C** in the attached PDF, and the results prove that the Opacity Penalty Strategy is indeed very effective, although it is simple. We will correct some places that are not clear enough in the revision.
>
> **_Q4: Advantages of sampling only in the horizontal direction._**
>
> Although some papers regard unseen views or adjacent training views as source views and warp them to the reference view, similar to our image-warping method based on binocular vision, these approaches do not consider the impact of the distance and angle between the source view and the reference view on the effectiveness of supervision. **Figure A** in the attached PDF shows the warped images and error maps when using different source views. It can be observed that when the source view is rotated relative to the reference view or is far away from the reference view, the warped image suffers from significant distortions due to depth errors and occlusions. This results in a large error compared with GT image, which hinders the convergence of the image-warping loss. In contrast, slight camera translations that cause minor changes in view without rotation and negligible impact from occlusions, allow the errors between the warped image and the GT image to primarily stem from depth errors, facilitating better optimization of depth.
>
> **Table A** in the attached PDF shows the quantitative comparison using different source images on the LLFF and the DTU dataset. The binocular vision-based view consistency method achieves the best performance.
>
> **_Q5: Using more diverse sampling space._**
>
> We perform ablation studies for horizontal sampling range $d_{max}$ on the LLFF and DTU datassets, as shown in **Table H** of the attached PDF, and the results show that the larger sampling range is not the better. There is almost no performance increase on both the LLFF and the DTU dataset when $d_{max}$ is greater than 0.4.
>
> Meanwhile, the **Table A** shows that the camera views with rotation lead to performance degradation.
>
> **_Q6: The impact of occlusion on performance._**
>
> Since we only move the camera slightly, there is no serious occlusion between the reference view and the source view, so we believe that the impact of occlusion can be ignored.
>
> **_Q7: An inaccuracy about the equation of image warping._**
>
> Thank you for pointing out the inaccuracy in the equation, warping the right image to the left view requires the depth of left image instead of the depth of right image. We will correct the equation in our revision.
>
> **_Q8: The impact of initialization on overall performance._**
>
> We conduct experiments on the LLFF and DTU datasets with several different initialization, including random initialization, sparse initialization, and using the LoFTR [1] matching network to construct initial point clouds. The results are shown in **Table K** of the attached PDF.
>
> PDCNet+ can get more matching points than LoFTR, so using the initialized point cloud obtained by PDCNet+ result in better performance. For the LLFF dataset, the input views have large overlapping, SfM methods can generate point clouds with better quality than LoFTR. Therefore, using sparse point clouds generated by SfM as initialization yields better performance than LoFTR. However, for the DTU dataset, where there is small overlapping among input views, SfM-generated point clouds are too sparse. So, the performance as initialization is even worse than random initialization.
>
> **_Q9: The results reported by FSGS differ significantly from those in the paper._**
>
> We cannot reproduce the results reported in the FSGS paper. The results listed in our paper are reproduced using their official code for a fair comparison. We will explain this in our revision. Note that even using the results reported in the FSGS paper, our results remain significantly better.
>
> **_Q10: Some visualization results are not labeled with the corresponding methods._**
>
> Due to some unknown reasons, the method names labeled in the visualization results cannot be displayed in some PDF readers. Please use Chrome to visualize these figures. We will address this issue in revision.
>
> **_Q11: Report the error bar or standard deviations._**
>
> We run our method 10 times for each scene  in LLFF and DTU dataset. Error bars are shown in **Figure D**.
>
> [1] LoFTR: Detector-free local feature matching with transformers. CVPR 2021.

---

> > ### Comment · Reviewer_gh5j · 2024-08-10
> >
> > Thanks for authors' response.
> > 1. I'm glad to see the ablation studies of the opacity penalty strategy. It would make the effectiveness of this module more convincing. But the motivation is still not clear for me. To my knowledge, this strategy adds a factor (less than 1.0) to weight down the original opacity and make a part of Gaussians below the pruning threshold, and how about directly adjusting the threshold of the pruning operation.
> > 2. The explanation of the binocular stereo consistency do not convince me enough. The authors explain that the advantage of only sampling the virtual view in the horizontal direction is the small rotation and less occlusion. And I think this is just the range control of the virtual view sampling space and can be achieved by any existing sampling methods, and don't show the advantage of the binocular sampling, after all, I can translate the camera in all direction but no rotation or small rotation.
> > 3. The results in Tab. K show that the dense matching model plays a very important role in the overall performance, and this method is relatively sensitive to the matching model. The effectiveness of the proposed modules seems have a great reliance of the matching model, and the random initialization performs poor especially on DTU dataset. And all these results should be analyzed and declared in the paper.
> > 4. Methodological errors and additional explanations about experiments and comparisons should be corrected and stated in the paper.
> >
> > Therefore, I still think my original rating is fair and this paper requires major revision on the method and experiment.

---

> > > ### Author Response · Authors · 2024-08-11
> > > **Responses to Reviewer gh5j**
> > >
> > > Ablation studies for the pruning threshold on LLFF dataset with 3 input views.
> > >
> > > | **pruning threshold** | 0.005 (baseline) | 0.010 | 0.015 | 0.020 | 0.025 | 0.030 | 0.035 | 0.040 | 0.045 | 0.050 |
> > > |-----------------|------------------|-------|-------|-------|-------|-------|-------|-------|-------|-------|
> > > | **PSNR**        | 20.48            | 20.30 | 20.23 | 20.05 | 19.93 | 19.93 | 20.09 | 19.96 | 20.08 | 19.91 |
> > > | **SSIM**        | 0.715            | 0.711 | 0.708 | 0.707 | 0.705 | 0.706 | 0.707 | 0.706 | 0.707 | 0.706 |
> > > | **LPIPS**       | 0.218            | 0.212 | 0.203 | 0.205 | 0.206 | 0.205 | 0.205 | 0.205 | 0.205 | 0.204 |
> > >
> > > We perform ablation studies for the pruning threshold, using dense initialization and view consistency loss by default. The experimental results show that directly increasing the pruning threshold leads to a performance decline.

---

> > > > ### Author Response · Authors · 2024-08-13
> > > >
> > > > Dear reviewer gh5j,
> > > >
> > > > As the reviewer-author discussion period is about to end, can you please let us know if our rebuttal addressed your concerns? If it is not the case, we are looking forward to taking the last minute to make further explanation and clarification.
> > > >
> > > > Thanks,
> > > >
> > > > The authors

---

> ### Author Response · Authors · 2024-08-11
> **Responses to Reviewer gh5j**
>
> Dear Reviewer gh5j,
>
> Thanks for  your feedback.
>
> **_Q1: How about directly adjusting the threshold of the pruning operation._**
>
> It is clear that the opacity penalty strategy is different from setting a fixed pruning threshold. The opacity penalty strategy is applied to all Gaussians and acts as a global constraint. In contrast, directly adjusting the pruning threshold only affects Gaussians with opacity lower than the threshold.  For Gaussians with opacity higher than the threshold but not on the scene surface, a fixed pruning threshold is ineffective.
>
> We are conducting experiments on the LLFF dataset with different pruning threshold, the results will be provided later.
>
> **_Q2: The range control of sampling space can be achieved by any existing sampling methods and the camera can be translated in all direction._**
>
> Yes, the existing methods might be able to change the position and angle of the unseen view to achieve the same effect as ours, but they didn't do it. To the best of our knowledge, our method is the first to attempt using viewpoint translation based on Gaussian Splatting to optimize the problem of sparse view synthesis, and it has been proven that it is more effective than other sampling methods.
>
> Indeed, the camera can be translated in any direction, but that still falls under translation without rotation, which we believe is essentially no different from horizontal movement.
>
> **_Q3: The method is relatively sensitive to the matching model and have a great reliance of the matching model._**
>
> Dense initialization is an indispensable part of our method, as we mentioned in Section 3.4 of our paper, our method needs a dense initialization to achieve better geometry initialization, thereby improving the quality of 3DGS. Additionally, experiments have shown that the matching model we selected is effective across different datasets.
>
> Although our method performs poor when using random initialization, it still outperforms the baseline 3DGS, especially on the DTU dataset, which proves that the other two components of our method are effective.

---

### Official Review · Reviewer_UVce · 2024-07-11

**Soundness:** 3
**Presentation:** 4
**Contribution:** 3
**Rating:** 7
**Confidence:** 4

**Summary:**

This paper proposes a new method related to the problem of novel view synthesis in a sparse input setting. The authors propose to exploit stereo consistency as a self-supervision signal in contrast to the use of priors such as diffusion which tends to produce less precise geometry. Specifically, the work proposes the use of binocular stereo consistency as a guiding signal, i.e. with a horizontal translation a disparity between two images can be calculated given intrinsics and depth, and this can be leveraged to warp the translated view to input image and assess the consistency between images. Furthermore, the authors address the issue of filtering the redundant gaussians by introducing a decay scheme based on penalising the opacity leading to pruning gaussians far from the surface of the object. The paper evaluates the proposed approach in a commonly used in literature sparse novel view scenario, using 3 common datasets. The authors present competitive results in their evaluation. An ablation of system elements/contributions is also included.

**Strengths:**

- This paper touches upon an interesting and important topic for the community - novel view synthesis in a constrained scenario. This work is among the first ones to propose a solution based on 3D Gaussian Splatting.
- The method proposed by the authors draws some inspiration from monocular depth estimation and formulates a self-supervision loss component applicable in 3D reconstruction. I find the idea novel and interesting - I appreciate the application of a relatively simple concept (correlation between depth and disparity) to the task.
- I believe that the big strength of the method is the focus on precise geometry and the lack of dependence on priors from pre-trained models.
- The paper clearly outlines performance improvements with respect to the previous state of the art, per-scene optimisation methods.
- The work evaluates the performance of the proposed method on 3 datasets well-known in the community showing improvements in all of them. I believe it is a suitable selection used throughout the literature (following the same setup of 3,6,9 views). The ablation study is suitable for the work showing the performance impact of separate components (dense initialisation, disparity consistency loss, opacity penalty).
- I find it particularly nice that the binocular stereo consistency can be added to effectively any 3D reconstruction method (provided having depth as an output). While I don't expect it to be heavily affecting dense reconstruction, I believe it would be looked into and tried to incorporate by many researchers working in sparse reconstruction.
- The paper is written clearly, I didn't have any trouble following it.
- The method is described in a way that should be reproducible - I believe I wouldn't have significant trouble reimplementing it.

**Weaknesses:**

- The main weakness I see is the lack of comparison to generalisable methods [1, 2, 3, 4, 5, 6, 7]. DTU dataset which is already used in the paper is also commonly used to benchmark methods that perform a training step on a selection of scenes. While the setup of generalisable, and per-scene optimisation is favourable towards the former, I believe the proposed method would be competitive. This would potentially be an interesting comparison, both quantitatively (I would expect competitive scores), and qualitatively (varying artefacts between the two approaches), and would additionally increase the value of the benchmark.
- I believe that computational performance comparison is a missed opportunity. Proposing one of the first methods for sparse novel view synthesis would benefit greatly by emphasising the speed differences between the methods. It would be a great addition to show the training and inference speed with respect to NeRF-based methods (to show improvement in quality metrics and speed), and other GS-based methods (to show probably similar speed, but increased reconstruction quality).
- Both disparity consistency loss and opacity-based pruning rely on the choice of hyperparameters - namely camera shift $d_{max}$ and decay coefficient $\lambda$. It would be great if this choice was motivated and its influence on the performance analysed (including analysis of whether the same values should be applied across tested datasets).
- The authors show estimated depth maps for their method with and without the use of view consistency loss. It would be good to see the depth maps from other methods (e.g. DNGaussian) - it would be interesting to see a comparison with methods that use depth priors as supervision. It would also be nice to see the comparison to ground truth depth (where not available, an interesting comparison would be the reconstruction method in a dense setup scenario - as a proxy for ground truth)
- The authors mention in the checklist that results in the paper are reported as an average performance - it would be good to specify whether experiments were run multiple times, and investigate how the performance varies across the runs.

[1] Alex Yu, Vickie Ye, Matthew Tancik, Angjoo Kanazawa, *pixelNeRF: Neural Radiance Fields from One or Few Images*, IEEE/CVF Conference on Computer Vision and Pattern Recognition, 2021

[2] Anpei Chen, Zexiang Xu, Fuqiang Zhao, Xiaoshuai Zhang, Fanbo Xiang
Jingyi Yu, Hao Su, *MVSNeRF: Fast Generalizable Radiance Field Reconstruction
from Multi-View Stereo*, IEEE/CVF International Conference on Computer Vision, 2021

[3] Mohammed Suhail, Carlos Esteves, Leonid Sigal, Ameesh Makadia, *Generalizable Patch-Based Neural Rendering*, European Conference on Computer Vision, 2022

[4] Haotong Lin, Sida Peng, Zhen Xu, Yunzhi Yan, Qing Shuai, Hujun Bao, Xiaowei Zhou, *Efficient Neural Radiance Fields for Interactive Free-viewpoint Video*, SIGGRAPH Asia, 2022

[5] Mukund Varma T, Peihao Wang, Xuxi Chen, Tianlong Chen, Subhashini Venugopalan, Zhangyang Wang, *Is Attention All That NeRF Needs?*, International Conference on Learning Representations, 2023

[6] Thomas Tanay, Matteo Maggioni, *Global Latent Neural Rendering*, IEEE/CVF Conference on Computer Vision and Pattern Recognition, 2024

[7] Haofei Xu, Anpei Chen, Yuedong Chen, Christos Sakaridis, Yulun Zhang, Marc Pollefeys, Andreas Geiger, Fisher Yu, *MuRF: Multi-Baseline Radiance Fields*, IEEE/CVF Conference on Computer Vision and Pattern Recognition, 2024

**Questions:**

- How was the camera shift $d_{max}$ and decay coefficient $\lambda$ chosen? Is the same value appropriate to all datasets? I would imagine it would be increased and decreased based on the distance of the object to the camera. Also, are the values of camera shift sampled uniformly?
- How would the application of disparity consistency loss to the existing methods look like? I would imagine it would be rather straightforward. Did the authors try that?
- Previous methods, as pointed out by the authors, use depth priors (using pre-trained models) as the supervision. Have the authors tried adding such supervision to their approach? I understand that the method is designed to be prior-free but it would be interesting to see an indication whether this would improve the performance.
- Zoom-in areas could be put side-by-side (possibly in the appendix, or another row in the figure with zoom-ins). It is a bit hard to compare small elements the authors want the reader to focus on when zooming the document - the focus areas (red boxes) are a bit far away and hard to notice small differences.

**Limitations:**

The authors provide a brief limitation section. I believe the issue with consistency constraints in textureless areas is a very good mention. Also, the experiment with masked training shows that the limitation is identified correctly.

---

> ### Author Rebuttal · Authors · 2024-08-07
>
> We sincerely appreciate the reviewer UVce for the invaluable feedback and time invested in evaluating our work. We respond to each question below.
>
> **_Q1: Comparison to generalizable methods._**
>
> Thank you for listing some of the latest generalizable novel view synthesis methods. We run the state-of-the-art MuRF method on the LLFF dataset for comparison with our method under the same input conditions. For the DTU dataset, we directly use the evaluation results from the original paper. As shown in **Table D** and **Table E** in the attached PDF.
>
> Our method outperforms the state-of-the-art MuRF method on the LLFF dataset, and also shows better performance when using 6 views and 9 views as input on the DTU dataset. MuRF performs better on the DTU dataset when using 3 views as input. However, generalizable methods require large scale datasets and time-consuming training to learn a prior.
>
> **Figure C** in the attached PDF shows the visual comparison between our method and MURF on LLFF dataset.
>
> **_Q2: Computational performance comparison._**
>
> Thank you for your advice. We provide a comparison on computational performance of our method and prior works. All the results are tested under one single RTX3090 GPU. The times are listed in **Table F** in the attached PDF.
>
> **_Q3: Ablation studies for hyperparameters $\lambda$ and_ $d_{max}$.**
>
> **Ablation study for $\lambda$.**
> We perform ablation studies on the LLFF dataset with the value of $\lambda$ ranging from 0.96 to 1.0, and the results are shown in **Table G** in the attached PDF.
>
> We find that the best performance is achieved when the value of $\lambda$ is 0.995. We set the $\lambda$ to 0.995 for all datasets.
>
> **Ablation study for $d_{max}$.**
> We perform ablation studies for $d_{max}$ on the LLFF and DTU datassets. The value $d_{max}$ gradually increases from 0.1 to 0.8, the results are shown in **Table H** in the attached PDF.
>
> We find that there is almost no performance increase on both the LLFF and the DTU dataset when $d_{max}$ is greater than 0.4, so we set the $d_{max}$ to 0.4 for all datasets.
>
>  The values of camera shift are sampled uniformly.
>
> **_Q4: Compare depth maps with other methods and pseudo-GT depth._**
>
> Figure 3 in our paper shows visual comparison of rendered novel images and depth maps between our method and others including DNGaussian. We apologize that due to unknown reasons, the method labels below the images cannot be displayed in some PDF readers. Please use Chrome to open our pdf to see the visual comparisons. We will address this issue in the revision.
>
> Additionally, the **Figure B** in the attached PDF presents comparisons of depth maps from several methods, including pseudo-GT depth. Pseudo-GT depth is obtained by training using all available views in baseline 3DGS.
>
> **_Q5: Experimental results from multiple run times._**
>
> We run our methods 10 times for each scene in LLFF and DTU dataset. Error bars are shown in **Figure D** of the attached PDF.
>
> **_Q6: Apply the disparity consistency loss to the existing methods._**
>
> We conduct experiments to train DNGaussian on LLFF dataset using disparity consistency loss. The results are shown in **Table I** of the attached PDF.
>
> Although we could not reproduce the performance reported in the original paper, we find that the performance is improved by comparing the quantification results before and after using disparity consistency loss. It indicates that disparity consistency loss is effective in other method.
>
> **_Q7: Apply depth priors as supervision._**
>
> We experiment with L1 depth loss and DNGaussian Depth Regularization on the LLFF and DTU datasets, employing dense initialization and opacity penalization by default. Monocular depth is obtained by pre-trained DPT, same as DNGaussian. We calibrate monocular depth using sparse point clouds from SfM when using L1 depth loss. The results are shown in **Table J** of the attached PDF.
>
> The L1 depth loss and DNGaussian Depth Regularization both result in performance degradation. We attribute this mainly to errors in monocular depth, even after calibration, leading to decreased performance.
>
> **_Q8: Put the zoom-in areas side-by-side._**
>
> We really appreciate your suggestion, and we will make appropriate adjustments to the layout of zoom-in areas in the revision.

---

> > ### Comment · Reviewer_UVce · 2024-08-11
> > **Thanks for detailed response**
> >
> > Dear Authors,
> >
> > Thank you for the amount of information provided in response to my review. I have read your rebuttal.
> >
> > I find the experiment with using your additional supervision with DNGaussian particularly convincing. I saw that performance is very competitive against MuRF (except 3 views DTU but that may be due to how DTU is evaluated, i.e. same views for all the scenes).
> >
> > One last worry I have is whether the choice of $\lambda$ is empirically easy. The method seems to be rather sensitive to small changes. Unless the value of $0.995$ is the best or close to best for all dataset.

---

> ### Author Response · Authors · 2024-08-12
> **Responses to Reviewer UVce**
>
> Dear Reviewer UVce,
>
> Thanks for your feedback.
>
> **_Q1: Whether the choice of $\lambda$ is empirically easy._**
>
> We conduct ablation studies for $\lambda$ on the DTU and Blender datasets, the results are as follows:
>
> Table G2: Ablation studies for $\lambda$ on DTU dataset with 3 input views.
>
> | $\lambda$ | 0.960 | 0.970 | 0.975 | 0.980 | 0.985 | 0.990 | 0.995 | 0.998 | 1.0   |
> |--------|-------|-------|-------|-------|-------|-------|-------|-------|-------|
> | PSNR   | 17.06 | 18.11 | 19.60 | 19.63 | 19.77 | 20.49 | 20.71 | 20.67 | 19.08 |
> | SSIM   | 0.785 | 0.813 | 0.846 | 0.848 | 0.853 | 0.861 | 0.862 | 0.862 | 0.832 |
> | LPIPS  | 0.223 | 0.196 | 0.158 | 0.152 | 0.139 | 0.121 | 0.111 | 0.109 | 0.131 |
>
> Table G3: Ablation studies for $\lambda$ on Blender dataset with 8 input views.
>
> | $\lambda$ | 0.960 | 0.970 | 0.975 | 0.980 | 0.985 | 0.990 | 0.995 | 0.998 | 1.0   |
> |--------|-------|-------|-------|-------|-------|-------|-------|-------|-------|
> | PSNR   | 23.68 | 24.24 | 24.37 | 24.48 | 24.57 | 24.69 | 24.71 | 24.59 | 23.47 |
> | SSIM   | 0.851 | 0.864 | 0.869 | 0.871 | 0.873 | 0.872 | 0.872 | 0.870 | 0.861 |
> | LPIPS  | 0.143 | 0.125 | 0.118 | 0.114 | 0.107 | 0.103 | 0.101 | 0.101 | 0.112 |
>
> We can see that the effect of $\lambda$ on performance is generally consistent across different datasets.

---

### Official Review · Reviewer_sng2 · 2024-07-12

**Soundness:** 2
**Presentation:** 2
**Contribution:** 3
**Rating:** 5
**Confidence:** 4

**Summary:**

This paper proposes 3D Gaussian splatting from sparse views aided by pre-trained key points matching initialization, binocular stereo constraints, and opacity regularization. Binocular stereo constraints utilize perspective projection to warp synthetic stereo images into the training images for self-supervision. Opacity regularization guides the densification/pruning process to drop non-active Gaussians.

**Strengths:**

Good initialization of Gaussians and geometrically inspired regularization for 3DGS (stereoscopic constrains and opacity decays)

**Weaknesses:**

1. Adding an opacity regularization does not seem enough to consider it a contribution.
2. Overstatements. The authors claim their method is prior-free, however, they use a very strong prior for dense initialization.
3. Unclear results. How is your method without any of your contributions (Table 4) 1.5 dB better than the baseline 3DGS?
4. Missing ablation studies. What is the performance if only stereo loss or only density regularization is applied?
5. I am afraid the main performance gains come from the dense keypoint initialization.
6. Inconsistent performance improvements in Table 4. Dense keypoint initialization provides ~3dB improvements, and stereo consistency and opacity regularization provides ~4dB. How is the combination of both only providing ~5dB? I understand improvements cannot be linearly added, but the gap seems unreasonable.

**Questions:**

See weaknesses

**Limitations:**

The authors provided limitations. However, the occlusions due to stereoscopic synthesis where not addressed (even though they are small due to small camera baselines).

---

> ### Author Rebuttal · Authors · 2024-08-07
>
> We sincerely appreciate the reviewer sng2 for the invaluable feedback and time invested in evaluating our work. We respond to each question below.
>
> **_Q1: Opacity regularization does not seem enough to consider it a contribution._**
>
> Although opacity regularization is a very simple strategy, it significantly contributes to improving the quality of sparse view synthesis. Similar to FreeNerf [1], its primary contribution lies in the use of frequency regularization, which despite its simplicity, greatly aids in performance enhancement. We provide more comprehensive ablation studies on the LLFF, DTU, and Blender datasets in **Table B** and **Table C** in the attached PDF. It can be observed that there is a significant performance improvement when opacity regularization is applied. We believe the simple yet effective regularization term will greatly benefit other researches on sparse view synthesis.
>
> **_Q2: Prior free claim._**
>
> Indeed, we did not describe 'Prior Free' clearly. What we intended to convey is that, different from methods such as the FSGS [2] and DNGaussian [3] that use priors as supervision, we do not require any prior as supervision, since we can mine supervision using the self-supervised method. We will correct this claim in the revision.
>
> **_Q3: 1.5dB higher than baseline 3DGS without any component used in Table 4._**
>
> Our method employs opacity penalty strategy, which conflicts with the opacity reset operation in baseline 3DGS, hence we do not use opacity reset operation. The first row in Table 4 shows the performance without using opacity reset operation. We inadvertently omitted an explanation of it in the ablation study, which we will add in the revision.
>
> **_Q4: Missing ablation studies._**
>
> We provide more comprehensive ablation studies on LLFF, DTU, and Blender datasets, as shown in the **Table B** and **Table C** in the attached PDF.
>
> **_Q5: The main performance gains come from the dense keypoint initialization._**
>
> Although dense initialization brings significant performance gains, relying solely on dense initialization cannot achieve or approach optimal performance. View consistency loss and opacity penalty strategy also contribute significantly to performance improvement, as evidenced on ablation studies in **Table B** and **Table C**  in the attached PDF..
>
> **_Q6: Inconsistent performance improvements in Table 4._**
>
> The three components we propose to improve the quality of novel view synthesis are essentially to make the Gaussians more accurately distributed on the scene surface, and the three components are complementary to each other. When only dense points are used as the initialization, the quality of Gaussians has a significant improvement space over the baseline 3DGS. Continuing with view consistency loss and opacity penalty further advances the already enhanced Gaussians, reducing the improvement potential. Therefore, there is less performance improvement when the three components are used together.
>
> Additionally, this is also verified by other works based on 3DGS, such as the ablation study in DNGaussian [3], which shows that using only Depth Regularization leads to a performance improvement of 2.5dB, whereas adding depth Normalization results in only a 1dB performance improvement.
>
> [1] Freenerf: Improving few-shot neural rendering with free frequency regularization. CVPR2023.
>
> [2] FSGS: Real-time few-shot view synthesis using gaussian splatting. arXiv preprint arXiv:2312.00451 (2023).
>
> [3] DNGaussian: Optimizing sparse-view 3d gaussian radiance fields with global-local depth normalization. CVPR2024.

---

> > ### Comment · Reviewer_sng2 · 2024-08-11
> > **Thanks for your reply**
> >
> > Thank you for addressing my comments. I am willing to raise the score, provided the clarifications and ablation studies are included in the final paper version.

---

> > > ### Author Response · Authors · 2024-08-11
> > > **Thanks to Reviewer sng2**
> > >
> > > Dear Reviewer sng2,
> > >
> > > Many thanks for all the helpful comments and positive assessment. We really appreciate your expertise and the score upgrade.
> > >
> > > Best,
> > >
> > > Authors

---

### Official Review · Reviewer_q3N3 · 2024-07-13

**Soundness:** 3
**Presentation:** 3
**Contribution:** 3
**Rating:** 5
**Confidence:** 4

**Summary:**

This paper introduces a novel method for 3D Gaussian-based sparse view synthesis. Specifically, initialized from dense point clouds, the depth-warping loss and  the opacity penalty strategy are introduced to obtain accurate 3D Gaussians. Extensive experiments on the Blender, LLFF and DTU dataset have demonstrated the effectiveness of the proposed method.

**Strengths:**

1. This paper is well-writen, the idea of depth-warping loss to improve view consistency is reasonable.
2. The proposed opacity penalty strategy is novel and does make sense. Through this simple operation, a better geometry can be obtained.
3. The experimental comparison is comprehensive.

**Weaknesses:**

1. Since depth-warping loss is common in many NeRF or 3D Gaussian-based papers, i'm not sure whether this can be viewed as a contribution.
2. Since the dense point clouds are generated from pretrained models, i do not think that this method can be claimed as prior free, please check the definition of what is prior free.

**Questions:**

Please see weaknesses. Since the initialization is based on dense point clouds, i think it would be better to do more ablation studies on more dataset, such as the DTU and the Blender dataset. However, in general, i think this is a relatively good work and i would recommond a borderline accept for this paper. I'm glad to improve rating if the weaknesses can be addrssed.

**Limitations:**

Please see weaknesses.

---

> ### Author Rebuttal · Authors · 2024-08-07
>
> We sincerely appreciate the reviewer q3N3 for the invaluable feedback and time invested in evaluating our work. We respond to each question below.
>
> **_Q1: Whether depth-warping loss can be viewed as a contribution_**
>
> Although some papers regard unseen views or adjacent training views as source views and warp them to the reference view, similar to our image-warping method based on binocular vision, these approaches do not consider the impact of the distance and angle between the source view and the reference view on the effectiveness of supervision. **Figure A** in the attached PDF shows the warped images and error maps when using different source views. It can be observed that when the source view is rotated relative to the reference view or is far away from the reference view, the warped image suffers from significant distortions due to depth errors and occlusions. This results in a large error compared with GT image, which hinders the convergence of the image-warping loss. In contrast, slight camera translations that cause minor changes in view without rotation and negligible impact from occlusions, allow the errors between the warped image and the GT image to primarily stem from depth errors, facilitating better optimization of depth. This is what we would like to report. We will highlight this in our revision.
>
> **Table A** in the  attached PDF shows the quantitative comparison using different source images on the LLFF and the DTU dataset. The binocular vision-based view consistency method achieves the best performance.
>
> **_Q2: Prior free claim_**
>
> Indeed, we did not describe 'Prior Free' clearly. What we intended to convey is that, different from methods such as the FSGS and DNGaussian that use priors as supervision, we do not require any prior as supervision, since we can mine supervision using the self-supervised method. We will correct this claim in the revision.
>
> **_Q3: Ablation study on DTU and Blender datasets_**
>
> Thank you for your suggestion, we provide ablation studies on DTU and Blender datasets with 3 input views. The results are shown in **Table B** and **Table C** in the attached PDF. We can see that our modules are effect on other datasets. We will update our ablation studies in our revision.

---

### Author Rebuttal · Authors · 2024-08-07

We sincerely thank the reviewers for their invaluable feedback and the time they dedicated to evaluating our work. We are pleased that the reviewers appreciated the representation and significance of the paper. We have addressed each reviewer’s comments separately, providing detailed analyses and ablation studies to resolve all the raised questions. The Visualization results and Tables are included in the attached PDF. Thank you once again for your insightful feedback, and we look forward to continuing the discussion.

---

### Author Response · Authors · 2024-08-08
**We will be happy to take any questions**

Dear reviewers,

We appreciate your comments and expertise. Please let us know if there is anything we can clarify further. We would be happy to take this opportunity to discuss with you.

Thanks,

The authors

---

### Decision · Program_Chairs · 2024-09-25

**Decision:**

Accept (poster)

**Comment:**

This paper received overall positive reviewers with engaged author and reviewer discussion after the author rebuttal.  The reviewers appreciated the novelty of the paper and the results.  The rebuttal and discussion addressed most concerns.

The remaining negative reviewers expressed the following concerns:

I mainly have three concerns: (1) there are methodological errors, e.g., the backward warping process. (2) the advantage of binocular sampling don't convince me enough, as the binocular sampling is a simplification of existing sampling method, and relative analysis is not appear in the paper. (3) from the results in Tab. K, it seems that the method is sensitive to the type of matching model, especially in DTU dataset, and I am not sure does this method have enough generalization.... I recommend the author including the experiments reported in the rebuttal, doing more sensitivity analysis about the type of matching models in different scenes, and giving more unique advantage of binocular sampling.

The AC reviewed all of this and this there is a nice contribution and that the remaining concerns can be addressed in the final revision.  Thus the AC recommends acceptance of the paper and that the authors very carefully include all experiments in the rebuttal and address the reviewers' comments.